# Visual impairment in pseudoexfoliation from four tertiary centres in India

**Aparna Rao**[ID][1]*, **Niranjan Raj**[1], **Amiya Pradhan**[1], **Sirisha Senthil**[2], **Chandra S. Garudadri**[2], **P. V. K. S Verma**[3], **Prakriti Gupta**[4]

**1** LV Prasad Eye Institute, Patia, Bhubaneswar, India, **2** LV Prasad Eye Institute, Hyderabad, Telangana, India, **3** Community Services, LV Prasad Eye Institute, Vishakapatnam and Hyderabad, Andhra Pradesh, India, **4** Glaucoma Service, LV Prasad Eye Institute, Vijayawada, Andhra Pradesh, India

* aparna@lvpei.org

## Abstract

### Purpose

To analyse the disease burden of pseudoexfoliation (PXF) disease stages from East and South India.

### Design

Prospective hospital based study of patients seen at 4 tertiary centres.

### Subjects, participants, and/or controls

Consecutive old and new patients of pseudoexfoliation with normal intraocular pressure (IOP), raised IOP (PXF with Ocular hypertension, OHT) and irreversible disc/field changes (pseudoexfoliation glaucoma, PXG) seen from April 2016-March 2017 at a tertiary centre in Odisha, East India and 3 centres in Andhra Pradesh and Telangana, South India, recruited into the prospective study were screened for baseline characteristics.

### Methods

The clinical and demographic details including visual acuity, laterality, intraocular pressure (IOP) with details of medical/surgical therapy at presentation were collected from the hospital database at all 4 centres.

### Intervention or exposure

The World Health Organization WHO visual criteria were used for defining visual impairment/absolute blindness in different disease stages.

### Outcome measures

The visual impairment/blindness rates with comorbidities in the anterior/posterior segment in PXF, OHT and PXG at baseline were compared and the influence of age, IOP fluctuations and laterality was analysed using multivariate logistic regression.

**Data Availability Statement:** All relevant data are within the paper and its Supporting Information files. We have provided raw data in submission.

**Funding:** This work was partly funded by the Wellcome Trust/DBT India Alliance grant: ref no IA/

CPHI/15/1/502031 awarded to Aparna Rao. All the funding or sources of support have been declared. There was no additional external funding received for this study.

**Competing interests:** The authors have declared that no competing interests exist.

**Abbreviations:** PXF, Pseudoexfoliation syndrome; PXG, Pseudoexfoliation glaucoma; IOP, intraocular pressure; XFM, exfoliative material; EMR, electronic medical record; TM, trabecular meshwork; OHT, Ocular hypertension; POAG, Primary open angle glaucoma; PACG, Primary angle closure glaucoma.

## Results

Of 6284 PXF eyes (of 3142 patients) included from all centres, OHT and PXG was seen in 2.1% and 29% respectively which included 3676 (>50%) bilateral PXF eyes. Reversible visual impairment rates caused by PXF associated co-morbidities in PXF and OHT were 33% and 26% respectively with cataract being the major cause (67% in PXF and 74% in OHT). Irreversible blindness rate was higher in bilateral PXG eyes (30.5%) compared to bilateral PXF (23.2%) or bilateral OHT (21.6%) with overall absolute blindness rates of 28.2% at presentation. Older age (p<0.001), bilaterality and higher baseline IOP were significantly associated with higher rates of blindness in PXF eyes.

## Conclusion and relevance

Pseudoexfoliation is associated with ≥30% visual impairment across all stages and 28% absolute blindness rate which is a huge hidden burden of glaucoma. Adequate disease staging and assessment of comorbidities is required for accurate prognostication at baseline and reducing avoidable pseudoexfoliation blindness.

## Introduction

The worldwide prevalence of pseudoexfoliation (PXF) and pseudoexfoliation glaucoma (PXG) varies widely ranging from 0–80% with maximal prevalence in Scandinavian countries. [1–4] Its unique clinical feature of flaky white pseudoexfoliative material (XFM) on different ocular structures combined with progressive neurodegeneration and systemic associations mandates its recognition as a unique identity among all forms of glaucoma. [5] Recognition of this entity is not only important to identify possible surgery related devastating complications in routine cataract surgery but also aids in identifying eyes at risk for developing long term irreversible ocular hypertension or glaucoma related blindness. [3,5–7] Early recognition and prognostication of this entity remains the only means for preventing onset of glaucoma since the pathogenesis of the irreversible glaucoma still remains a mystery. [1] Details on the vast clinical spectrum of the disease and the frequency of blindness due to PXF at presentation are scarce in literature with most studies focussing on genetic aetiology, pathogenesis or prevalence of the disease in a particular geographical location or country. [5,8,9,10]

The long term progression from glaucoma suspect to ocular hypertension or established glaucoma is well characterised in randomised controlled trials for primary open angle glaucoma (POAG) and primary angle closure glaucoma (PACG). [11,12] Yet parallel epidemiological studies on pseudoexfoliation focus mostly on prevalence rates in specific geographical locations with minimal or no reference to its different stages or blindness rates in different disease stages like PXF only, ocular hypertension (OHT) and pseudoexfoliation glaucoma (PXG). [8–10] This disease is very well characterised clinically with detailed description of the complications associated with cataract surgery performed in eyes with PXF. [1,5,6] Also known is the faster rate of visual field progression in eyes with PXF which is partly explained by refractoriness to conventional therapy in these eyes. Very few studies have described the extent of visual impairment of the disease stages at baseline which may possibly give a clue to the actual disease burden of the disease across its varied stages. [6] This information is crucial to prognosticate an eye with PXF to prevent disease progression and irreversible damage over long term. This risk also needs to be compared across geographical locations to compare the effect of

environment on baseline risks since PXF is known to be highly influenced by climatic and environmental conditions. We had earlier described varied phenotypes and stages of this entity seen in east India and identified increased risk in specific phenotypes of the disease. [5,6] This study evaluates the disease burden and blindness rates across different PXF stages in patients seen at 4 tertiary centres from two geographically different locations/states with different climatic profiles of India(https://www.mapsofindia.com/maps/india/annualtemperature.html).

## Methods

Hospital record database at 4 tertiary eye care centres in South (n = 3) and East (n = 1) India of consecutive new and old cases of pseudoexfoliation seen during April 2016 to March 2017 were recruited for a prospective study and screened for baseline variables. The method for ophthalmic work up, stratifying PXF cases into phenotypes or stages and data retrieval from electronic medical record database is detailed elsewhere. [5,6] In brief, the institutional electronic medical record (EMR) captures all data including demographic, clinical details including Goldmann applanation intraocular pressure (IOP), slit lamp findings, gonioscopy, fundus biomicroscopy findings, medications, surgeries and visual fields data. These data were retrieved from different sections of the patient record in the EMR completed by different perimetrists, optometrists or ophthalmologists and stored in the central EMR server. This study was approved by the institutional review board of each centre including LV Prasad Eye Institute, Mithu tulsi chenrai (MTC) campus Bhubaneswar, KAR campus Hyderabad, KVC campus Vijayawada and GMRV campus Vishakapatnam and the study adhered to the tenets of declaration of Helsinki. Details including demographic information of each patient, slit lamp evaluation with slit lamp photographs, best corrected Snellen visual acuity, clinical pattern of PXF deposits (if described),+90D fundus biomicroscopy and Humphrey visual fields (Carl Zeiss Inc, Dublin, CA, USA, 24–2 SITA standard program), were retrieved from the EMR database at baseline and every visit till final follow up. We also collected details of referrals in the database (which is captured in the clinical EMR database for patients which have been referred to us for tertiary care as opposed to those who presented primarily to us), pattern of deposits, diagnosis and other ocular associations. A uniform definition of PXF and its variants or stages were used to compare data as described elsewhere and below in brief (See below for details of clinical definition or stratification). [5,6] Patients with incomplete data or indeterminate diagnosis of pseudoexfoliation due to pupillary deposits (for example in uveitis) were excluded. Our earlier study based on a large sample over 2 years found a prevalence rate which was comparable to the disease rates in India and across the globe suggesting that our sample was a close representation of the population sample in our region. [6,7] Recruitments were done between April 2016 to March 2017 for patients fulfilling inclusion criteria as defined below and a written informed consent from each patient was obtained as per institute protocol.

Management of these cases at each centre including anti-glaucoma medications at baseline and each visit was done as per the clinical faculty's discretion. Decision for surgery in each case and referral to other departments was carried out as per clinical requirements and details of surgery with intraoperative complications (if any) and postoperative course till final follow up were noted for all patients from each location from the central database server. The surgical outcomes at 1 year from each centre were also compared.

## Clinical definitions and stratification

The definition of Pseudoexfoliation and the clinical variants has been described earlier. [5,6] Briefly, classical pseudoexfoliation was diagnosed in eyes with evident flaky classical dandruff

like exfoliative deposits on the pupil, lens or other ocular structures, open or closed angles on gonioscopy with or without radial pigment over the lens surface on dilated slit lamp evaluation. The three clinical phenotypes based on the clinical features have been described in our earlier study. [5] Eyes with clinically evident PXF in any one eye and normal IOP/visual field and optic nerve were classified as pseudoexfoliation syndrome while those with raised IOP >21mm Hg, normal optic nerve and visual field were diagnosed as PXF with OHT after exclusion of other types of glaucoma. Pseudoexfoliation glaucoma (PXG) was diagnosed in those with glaucomatous optic neuropathy evidenced by cupping, focal notch or retinal nerve fibre layer defects with corresponding reproducible glaucomatous visual field defects (defined as glaucoma hemifield test outside normal limits or pattern standard deviation with probability <5% with fixation losses<15% and false positives and false negatives <30% determining reliability criteria). For those whose fields could not be performed or were unreliable owing to significant cataract/other media opacities, presence of glaucoma was ascertained after cataract removal by evidence of structural glaucomatous damage to the optic nerve.

Unilateral and bilateral eyes and eyes with prior filtering surgeries, lasers, medications or other interventions for controlling IOP were included in the presence of exfoliation material in the eye while absence of clinically evident PXF at any visit (in the contralateral eye of a patient with unilateral PXF) was considered as clinically "normal" eye when associated with a normal IOP and disc/posterior segment. Patients with any other autoimmune or neurodegenerative disorder with corresponding disc changes were excluded. Associated diagnosis noted by clinician in the contralateral eye of unilateral PXF or same eye like lens induced phacomorphic/phacolytic or secondary lens induced angle closure glaucoma associated with the presence of PXF were noted. Co-morbidities in the anterior or posterior segment were recorded and analysed. Medical management was initiated for adequate IOP control in all visits with addition of anti-glaucoma medications while decision for additions/switch or stoppage of medicines (for totally blind asymptomatic eyes) was decided by the clinician at each visit. Follow up management of all eyes with definition of target IOP and surgical success or failure was noted by surgeons at each centre as per standard criteria for every patient.

While blindness and visual impairment definitions vary across studies, measurement of visual field and best corrected visual acuity (BCVA) is not possible for most studies. Since reliable visual field is seldom obtained at the first visit in these PXF eyes with visually significant cataract, the visual criteria henceforth are used for defining disease burden in PXF. So this study herein defines severe visual impairment as measured central visual acuity of 20/200 or less with the best possible correction with/or without a reliable visual field of 20 degrees or less. Eyes with absolutely no measurable amount of useful vision (projection of rays/perception of light) or Snellen BCVA<20/400 which was reproducible at all visits were termed as absolute blindness. Eyes with fluctuating vision between two consecutive visits were noted separately and not included in the main analysis. The eyes were stratified into 4 groups based on best corrected visual acuity at baseline and final visit as follows- 0-absolute blindness as defined above; 1-<20/200 and />20/400; 2-<20/100 and >20/200; 3-<20/40 an >20/100 and 4->20/40.

## Statistics

All data were pooled to a common server for analysis using Stata corp (USA, Version 10) with alpha error set at p<0.05. Based on our earlier study on >11,000 cases of PXF, we found a prevalence of 11.5% which was comparable to population based studies in India and across the globe. [5,6,7] The sample size required for the study was calculated to be 155 with a projected prevalence of 11.5%, precision set at 0.05 and 95% confidence intervals. Since we were studying different geographical locations, we chose a minimum sample of 200 from each centre

though we included all patients fulfilling inclusion criteria seen and recruited during the study period. Continuous variables are represented as mean (±standard deviation, SD) or median (range) while frequency of features are represented as numbers (n,%). The relative proportion of visual impairment/blindness was analysed separately in each group like PXF stages or unilateral and bilateral PXF while overall proportions among all patients were also collected at baseline. Though there is an increased risk of having pseudoexfoliation in an unaffected eye in unilateral disease, this is highly variable with no clear proven correlation between the two eyes of the same patient. [1] Since the diagnosis or severity of PXF in one eye has no clear proven correlation to the risk /condition of the other eye, each eye was taken as a separate unit with no statistical adjustment made for possible correlation between the two eyes of the same patient. The difference in demographic and clinical variables across different centres and different groups was compared using ANOVA statistics. The rates of OHT and PXG and rates of severe visual impairment /blindness across all geographical areas were also compared while proportion of OHT/PXG among cohort in different locations were compared using Chi square. Logistic regression was used to analyse the influence of independent actors like age, IOP, sex, PXF stage, laterality or visual field parameters on risk of blindness.

## Results

The mean age of 6284 eyes of 3142 patients seen at all 4 centres during the recruitment period, was 67±8.8 years which included majority of males (F:M = 33%: 67%). Among these, 6.2% were referred patients while the rest presented primarily to the respective centres for the first ophthalmic consultation. This comprised of 1304 unilateral PXF (41.5%) and 1838 (58.5%) bilateral PXF disease. Among 1304 unilateral PXF patients, the contralateral eye was diagnosed as clinically normal at the time of presentation in 1216 patients while 88 were noted by the clinician to have other forms of secondary lens induced glaucoma, Table 1.

Stratifying the eyes according to PXF stages, 3024 eyes were noted be having PXF with no evidence of raised IOP or disc damage, 144 OHT eyes with raised IOP and no disc/field

**Table 1. Demographic and clinical characteristics of eyes with pseudoexfoliation of different stages across 4 tertiary centres and two geographic locations in East and South India.**

| | PXF N = 3024 | PXF with OHT N = 144 | PXG eyes* N = 1804 | Clinically normal eyes N = 1216 | P value^ |
|---|---|---|---|---|---|
| Age (years) | 67±8.7 | 66±8.6 | 74±9.2 | 65±7.9 | 0.04 |
| Unilateral: Bilateral (n, %) | 961:2063 31.7%:68.2% | 21:123 14.5%:85.5% | 322:1482 17.8%:82.2% | NA | NA |
| Baseline IOP(mm Hg) | 15±5.02 | 18±6.1 | 24±11.5 | 14±4.8 | <0.001 |
| SD of IOP over visits (range, mm Hg) | 7.5(3.1–9.4) | 10.3(4.1–12.3) | 9.5(6–28.4)$ | 3.6(1.4–6.6) | 0.002 |
| Number of AGM at presentation (n,%) | 1–108 (3.5%) | 1-97(67.3%) | 1-323(17.9%) | 1-46(37.8%) | <0.001 |
| | 2-92(3.04%) | 2-16(11.1%) | 2-559(30.9%) | 2-34(28%) | <0.001 |
| | 3-38(1.2%) | 3-5(3.4%) | 3–376 (20.8%) | 3-29(23.8%) | <0.001 |
| | 4-8(0.2%) | | 4–104 (5.7%) | 4-11(9.5%) | <0.001 |
| | 5–3 (0.09%) | | 5-12(0.6%) | | <0.001 |
| Time of surgery (months) | 33±24.7 | 31±21.7 | 32±17.5 | 34±31.9 | 0.09 |
| Baseline Legal blindness rate#(n,%) | 992 | 38 | 632 | NA | |
| | 32.8% | 26.4% | 35% | | |

*8 eyes with variable indeterminate vision are not included here

#-See text for full definition of legal blindness

$ Including some eyes with development of end stage glaucoma(neovascular glaucoma for example)

^- One Way Anova with maximum differences on posthoc analysis between controls/PXF with PXG; SD-standard deviation; IOP-intraocular pressure; AGM-anti-glaucoma medications; PXF- Pseudoexfoliation; OHT-Ocular hypertension;: PXG-Pseudoexfoliation glaucoma

damage,1804 PXG with established glaucomatous changes, **Table 1**. Age at presentation was greater for males (74±4.9 years) than females (65±8.8 years) across all geographic regions, p = 0.03. The OHT and PXG eyes presented later than PXF, p = 0.04, **Table 1**. The proportion of OHT or PXG was not significantly different between the two locations of East and South India.

The mean baseline IOP of all eyes was 33±24.7mm Hg with highest IOP seen in PXG eyes, p<0.001, **Table 1.** Of 1804 PXG eyes, >50% of eyes required more than 1 medicine with the number of PXG eyes requiring >3 or 4 medicines being significantly higher than PXF or OHT eyes. The OHT eyes required 1 medicine in 67% while only <5% eyes required>2meds for IOP control. It was surprisingly noted that 103 eyes of 1216 normal contralateral eyes of patients with unilateral PXF were receiving medicines prescribed for the eye with PXF at presentation (**See Table 1**). Control of IOP with medicines was achieved by the third visit across all categories with appropriate medical/surgical treatment (mean follow up of 6±3.2 months) with >50% eyes requiring medicines in PXG eyes and OHT eyes for smoothening IOP fluctuations at follow up visits, **Table 1**. The mean fluctuation of IOP in the study period of 1 year at different visits ranged from 4mm-15mm Hg in PXG eyes while that for OHT and PXF eyes was minimal, **Tables 1 and 2**.

Comparing unilateral and bilateral PXF eyes, bilateral PXF eyes had significantly higher IOP than unilateral PXF eyes even at 3rd visit and at 1 year with higher SD in the former**, Table 2**. More number of bilateral PXG eyes (52%) required >2 medicines compared to unilateral PXG eyes (28%) suggesting a severe clinical course at presentation in bilateral PXG eyes. This however did not reflect in overall higher total requirement of anti-glaucoma medications (AGM's) in bilateral PXG suggesting that most bilateral PXG eyes that were blind at presentation may not have been started on medical treatment owing to nil visual prognosis, **Tables 1 and 2**.

## A. Visual impairment and blindness rates in PXF stages

The clinical profile and visual acuity in all stages of PXF as also bilateral and unilateral cases is shown in **Tables 2–4 and S1 Table**. Unilateral cases had <20% of eyes with visual impairment in all stages at presentation which was significantly lower than bilateral eyes at presentation, **S1 Table and Table 3**.

Comparing stages of glaucoma, PXG eyes had highest rates of visual impairment and absolute blindness at baseline and 1 year, p<0.001. Eyes with bilateral disease had greater rates of

**Table 2. Clinical profile of unilateral and bilateral pseudoexfoliation eyes across 4 tertiary centres.**

| Variables | Unilateral N = 1304 | Bilateral N = 3676 | P value* |
|---|---|---|---|
| Age (years) at presentation | 66±8.5 | 68±9.06 | 0.058 |
| Male:Female (%) | 64.3:35.6 | 64.1:35.9 | 0.4 |
| IOP baseline(mm Hg) | 17±9.1 | 21±9.2 | |
| Final IOP (mm Hg) | 15±5.6 | 14±1.9 | 0.8 |
| SD of IOP (range, mm Hg) | 3.6 (10.8–12.6) | 8.06 (11.8–28.4) | 0.02 |
| No of AGM | 1–135 (10.3%) | 1-336(9.1%) | 0.6 |
| | 2–149 (11.4%) | 2-427(11.6%) | 0.9 |
| | 3-94(7.2%) | 3-251(6.8%) | 0.4 |
| | 4–20 (1.5%) | 4-74(2.01%) | 0.1 |
| | 5–5 (0.3%) | 5-10(0.2%) | 0.8 |

*- student t test; IOP-intraocular pressure; AGM-anti-glaucoma medications; SD-standard deviation

**Table 3. Relative proportion of visual impairment or absolute blindness rates in a hospital based multicentric cohort of PXF (Pseudoexfoliation), PXG (Pseudoexfoliative glaucoma), and OHT (PXF with ocular hypertension) at baseline and final visit.**

| Baseline | PXF N = 3024% | OHT N = 144% | PXG N = 1804% | P value | 1year | PXF N = 3024% | OHT N = 144% | PXG N = 1804% | P value |
|---|---|---|---|---|---|---|---|---|---|
| Severe Visual impairment* | (32.8) | (26.4) | (35) | | Severe Visual impairment | (25.4) | (20.1) | (44.6) | |
| Absolute *blindness | (22.2) | (21.5) | (25.6) | | Absolute blindness | (16.6) | (16) | (22.8) | |

*-refer to text for detailed summary of visual impairment/ blindness definitions PXF- Pseudoexfoliation; OHT-Ocular hypertension; PXG-Pseudoexfoliation glaucoma

visual impairment (40.6%) compared to unilateral disease (11%) even at presentation, p<0.001, **Table 2 and S1 Table, Fig 1**. In each group, the relative proportion of visually impaired eyes at end of study period at 1 year after appropriate medical/surgical interventions was statistically higher (14.6%) in bilateral PXG eyes compared to bilateral PXF (10.3%) or bilateral OHT (8.8%) or unilateral PXF of any stage, **S1 Table**. A worse prognosis in bilateral PXF is suggested by 16% of absolutely blind eyes at baseline and 14% at 1 year after appropriate clinical interventions, **Fig 1**. This may be due to slightly delayed presentation at baseline in bilateral PXF compared to unilateral PXF, p = 0.058 and a higher proportion of irreversible absolutely blind eyes at presentation, **Tables 2 and 3 and S1 Table**. This translated to a >32.9% of overall combined visual impairment and 22.8% absolute blindness in all eyes with pseudoexfoliation in this hospital based cohort at baseline which did not change at 1 year (26.8% and 18.5%, respectively) after standard medical/surgical treatment in all eyes, **Tables 2 and 3**.

The causes of visual impairment in different stages of PXF at baseline are detailed in **Table 4**. Excluding the reversible causes due to non-operated visually significant cataract which formed the major cause (66–74% across stages), the burden of severe visual impairment and absolute blindness due to irreversible PXG in unilateral and bilateral cases still remained high with significantly higher blindness rates due to PXG in East India compared to South India, **Fig 2**. Both posterior and anterior segment co-morbidities were associated with PXF related visual impairment in 10–25% of eyes with posterior segment pathologies predominating across PXF stages, **Table 4**.

## Secondary glaucoma in PXF

Pseudoexfoliation is known to be associated with lens induced glaucoma owing to frequent subluxation of the lens and zonular laxity. In eyes with secondary forms of glaucoma (88 eyes

**Table 4. Causes of blindness among all stages of pseudoexfoliation patients seen at two geographical areas.**

| Stage of disease | Cause of blindness/visual impairment |
|---|---|
| PXF | Un-operated Total/near total Cataract (67%), Posterior segment pathology (17.1%)-Disc pallor, Old Vein occlusion with or without maculopathy Anterior segment pathology (13.2%)- Corneal ulcer, opacities, microbial keratitis, Others (5.7%)-eg. Old retinal detachments or other pathologies like traumatic optic neuropathy |
| OHT | Un-operated Total/near total Cataract -74.3% Posterior segment pathology (25.7%)- Vein occlusion, Disc pallor |
| PXG | Associated Cataract-66% Posterior segment (23.2%)-Glaucomatous optic neuropathy, Endophthalmitis, Disc pallor, Maculopathy after VR surgery Anterior segment pathology (10.8%) Corneal opacities |

PXF-Pseudoexfoliation; OHT-PX PXF with ocular hypertension; PXG-Pseudoexfoliation glaucoma

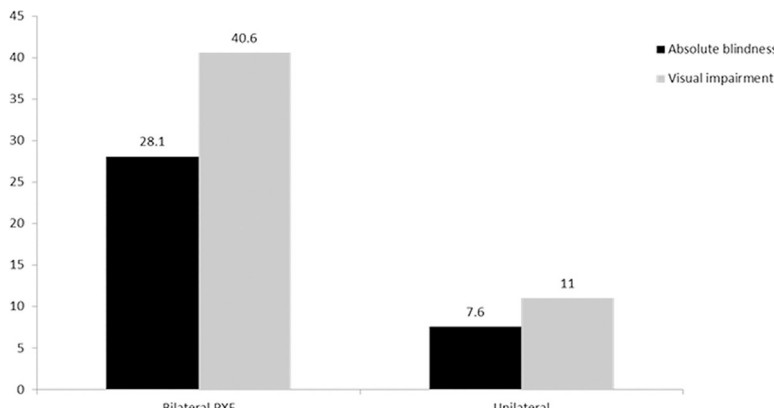

**Fig 1. Overall rate of visual impairment and absolute blindness in unilateral and bilateral pseudoexfoliation at baseline showing significantly higher numbers in eyes with bilateral disease (refer to methods for definition of absolute blindness).**

**Fig 2. Relative proportion of visual impairment and absolute blindness at baseline and final visits between two geographical locations of East and South India (see text for definitions of visual impairment and absolute blindness, asterix indicates comparison rates between East and South India which were significantly different or p<0.05).**

with lens induced glaucoma in the contralateral normal eye of unilateral PXF and 8 eyes of unilateral PXF), 55 eyes were on medical treatment while the rest had IOP control after Nd: YAG iridotomy for gonioscopic evidence of angle closure. These secondary glaucoma eyes had a baseline IOP of 20±12mm Hg with final IOP of 15±2.1 mm Hg at 1 year, **S2 Table.** The secondary glaucoma eyes had 35% visual impairment and 28% of absolute blindness at presentation which was significantly higher than the blindness rates in other PXF stages, $p < 0.001$, **S2 Table** and **Table 1**.

The proportion of visual impairment and absolute blindness may be influenced by clinical variables/presentation on the disease severity. To evaluate factors influencing PXF related vision loss at baseline, multivariate analysis was done with independent variables like age, laterality, sex, disease stage, baseline IOP, IOP fluctuations (SD) and baseline visual field parameters like visual field index and pattern standard deviation. Logistic regression showed older age, bilaterality and higher baseline IOP to be significantly associated with higher rates of blindness in PXF eyes, **S3 Table**.

## B. Surgery outcomes in PXF cases

We followed up cases which underwent surgery for associated cataract or co-morbidities and evaluated the outcome at the end of the study period from baseline. Of all cases, pupillary dilatation<5mm was seen in 27.8% eyes while rest had good dilatation with >6mm dilatation in 2738 eyes (56.7%). **S4 Table** details the type of surgeries performed in eyes with PXF with causes for posterior segment associations. Six eyes had intraoperative complications of zonular dialysis. Many eyes could undergo successful cataract or combined cataract and glaucoma surgery with good functional central visual acuity at final visit (visual gain >2 snellen lines in 86% of eyes undergoing surgery) with no intraoperative complications. While total number of eyes with visual acuity 20/40 or more were significantly lower at final follow up for bilateral cases compared to unilateral PXF, unilateral or bilateral cases undergoing surgery had comparable visual outcomes with no postoperative loss of vision seen in any group. Transient rise of IOP immediately after surgery resolved at 1 week follow up in all cases with none requiring additional medicines/intervention at 1month after surgery. One eye with severe endophthamitis (unknown aetiology) underwent evisceration while 34 eyes with associated lacrimal problems required lacrimal surgery. Postoperative outcomes of trabeculectomy combined with phacoemulsification or small incision cataract surgery were comparable with no statistical difference between final IOP or best corrected visual acuity between the two surgical techniques.

## Discussion

This study found relatively high rate of irreversible (and reversible) absolute blindness and visual impiarment in this hospital based cohort spanning two geographical locations. The rates of visual impairment/blindness were significantly higher for bilateral PXG eyes than unilateral PXG or OHT. While>2/3rd cases of OHT were adequately controlled with 1 medicines, PXG eyes required more than 2 medicines for IOP control. The bilateral cases presented later than unilateral cases, required more medications at presentation and also had greater fluctuations of IOP compared to unilateral cases though the number of AGM was not significantly higher than unilateral cases owing to higher rates of total blindness in the former. Secondary glaucoma is not infrequent in these eyes at presentation where the blindness rates are higher than in PXF eyes with primary PXG. Surgeries for visually significant cataract or uncontrolled IOP in these cases are associated with good visual outcomes. With proper preoperative and intraoperative precautions, complications can be significantly decreased in eyes with secondary glaucoma though the final visual outcome may be poor owing to advanced glaucoma.

Blindness rates due to glaucoma differ significantly across various regions of the globe ranging from 11–30% in different types of primary glaucoma, including POAG and PACG. [12–16] Similar studies on PXF report different prevalence rates and rates of PXG associated blindness. [1–3,5,11,12,17,18] Unilateral and bilateral blindness rates in PXF across 7 glaucoma services report a lower prevalence of <15% and 1.2 respectively which was in contrast to our findings. [17] A population based study in southern India reported a rate of 20.5% blindness in PXF with unilateral blindness accounting for 40% though this study did not stratify patients into different stages nor did it reflect the actual burden of reversible versus irreversible blindness.[9] In another population based study with 25% of bilateral PXF blindness, 89% was due to cataract thereby suggesting that the actual irreversible blindness rates may be lower in that population as compared to our hospital based cohort. [13] The very high rates of blindness in this study may reflect a more delayed presentation leading to higher rates of irreversible blindness reflecting the loomingly huge burden of reversible and irreversible blindness/visual impairment rates in PXF compared to other types of glaucoma. The differences in rates of visual impairment across studies arise due to difference in defining criteria for blindness (VA or VF) along with lack of stratification of PXF into different stages as in this study. Our earlier study reported prevalence rates of different stages of PXF at one centre in eastern India with higher prevalence of OHT(20%) and PXG(50%) in bilateral as compared to unilateral PXF(12 and 25% respectively). [5,6] Since PXF is known to have environmental influences on aetiology and prevalence, we chose two different geographical locations with different profiles; [1,2,3] yet we found minimal differences in proportion of OHT/PXG across the two states suggesting that the risk of stage transitions in PXF were minimally affected by climatic related triggering factors. Baseline difference in PXF prevalence across the globe may possibly only define the baseline genetic predisposition to developing PXF disease and thereby the baseline prevalence rates in different ethnic populations while not particularly affecting the risk for developing glaucoma or OHT. This theory requires concerted large scale efforts at genotype-phenotype correlations with accurate stratification of PXF stages. POAG is recognised as the major cause for blindness in most countries while PACG accounts for majority in select countries with high prevalence of angle closure disease. [11,12,14,15] This study identified higher rates of total blindness and visual impairment in bilateral PXG which is important for formulating policies for reducing PXF blindness and spearheading programs for better estimation of PXF burden in the community in addition to common forms of primary glaucoma.

Earlier studies have reported higher rates of PXF associated blindness dependent on age and baseline IOP though they have not analysed the burden of visual impairment across laterality and PXF stages. [1,8,10,18] This study observed an aggressive course in bilateral PXG patients with higher IOP in bilateral than unilateral cases. Clinical differences between unilateral and bilateral cases are known in PXF with conversion to the latter over time representing that bilateral disease may be a later clinical form of unilateral PXF. [6,8,10]' The nerve fibre layer in unilateral cases may be thinner compared to normals thereby representing that the manifest form of disease may be considered to be a stage in transition to more aggressive bilateral form over time. [19] Yet, the predictor for risk of conversion to bilateral cases or to PXG in any eye with PXF is largely unclear with no correlation to clinical findings. [1,5] Understanding the genetic landscape in each may suggest if genetic susceptibility also determines an aggressive course in bilateral cases as compared to unilateral cases. Nevertheless, identifying the stage of any PXF eye definitely helps clinicians to predict and prognosticate a possibly faster progression in bilateral PXF as seen in this study to enable timely and aggressive management in these eyes.

We found comparable visual outcomes after surgery in eyes with unilateral and bilateral PXF and found good visual outcomes after surgery in majority of cases. Surgical outcomes in PXF and associated cataract is associated with intraoperative vitreous loss, zonular dialysis,

dropped nucleus and postoperative rise in IOP among an array of complications arising due to disease pathophysiology affecting different ocular structures. [20–24] While uncomplicated PXF with cataract has comparable surgical outcomes with those of non-PXF eyes, PXF eyes associated with miotic non-dilating pupils, hard nucleus, pre-operative zonular laxity and associated glaucoma pose a significant challenge to surgeons. [7,22,24–26] While most studies focus on phacoemulsification technique in PXF, prospective studies on outcomes with small incision surgery for cataract or combined procedures for glaucoma, most commonly practiced technique in developing countries, versus phaco techniques in these eyes are scarce or absent. [25,26] Usually, these eyes require sphincterotomy or pupillary dilatation with iris hooks intraoperatively. [23] This again causes more trauma as well as heightened inflammatory post-operative inflammation. This study did not find miotic non dilating pupils even in eyes with evident PXF suggesting that most of these eyes can be well managed surgically if adequate pupillary details and lens grade are noted preoperatively. Many eyes underwent lacrimal surgeries as well as retinal procedures mandating careful preoperative evaluation to rule out these associations in these eyes.

The strengths of our study included the large dataset carried out across two geographical locations to explore any effect of environment on disease burden and blindness rates across PXF stages. This was a prospective study thereby reducing inherent flaws of a retrospective design. Further, we used careful and stringent stratification of the stages of PXF to know the risk of blindness in each stage which is crucial and important for accurate prognostication for each eye in glaucoma practice. We also used WHO blindness visual acuity criteria which reflected the true burden of PXF disease in our population which would possibly be used for driving health care planning strategies or avoiding PXF blindness in the two states. Though our earlier hospital based study showed prevalence rates of PXF comparable to population based studies, this was a hospital based prospective study involving 4 tertiary centres; so rates of OHT and PXG or blindness rates in this hospital based cohort may not reflect or may differ from the true population based prevalence rates across the two geographic locations to some extent. This was a multicentric study involving multiple trained surgeons; yet we do not believe this caused any difference in medical or surgical management outcomes in PXF in view of standardised protocols followed at all centres.

## Supporting information

**S1 Table. Baseline visual acuity and final visual acuity of patients with unilateral or bilateral pseudoexfoliation at different stages (see methods for detailed description of visual acuity groups 0–4).**
(DOCX)

**S2 Table. Characteristics of eyes with pseudoexfoliation and secondary glaucoma across 4 tertiary centres.**
(DOCX)

**S3 Table. Factors influencing higher risk of blindness in patients with pseudoexfoliation.**
(DOCX)

**S4 Table. Surgeries done with complications and final outcome in patients with pseudoexfoliation in 4 tertiary centres.**
(DOCX)

**S1 Data.**
(XLS)

## Acknowledgments

Electronic medical records institute team of LV Prasad Eye Institute which has been instrumental for collecting details and data from across all campuses.

## Author Contributions

**Conceptualization:** Aparna Rao.

**Data curation:** Aparna Rao, Niranjan Raj, Amiya Pradhan, Sirisha Senthil, Chandra S. Garudadri.

**Formal analysis:** Aparna Rao, Niranjan Raj, Amiya Pradhan, Sirisha Senthil, Chandra S. Garudadri, Prakriti Gupta.

**Funding acquisition:** Aparna Rao.

**Investigation:** Aparna Rao, Niranjan Raj, Amiya Pradhan, Sirisha Senthil, Chandra S. Garudadri.

**Methodology:** Aparna Rao, Amiya Pradhan, Sirisha Senthil, Chandra S. Garudadri, P. V. K. S Verma, Prakriti Gupta.

**Project administration:** Aparna Rao, Niranjan Raj, Amiya Pradhan, Sirisha Senthil.

**Resources:** Aparna Rao, Sirisha Senthil.

**Software:** Aparna Rao.

**Supervision:** Aparna Rao, Niranjan Raj, Amiya Pradhan, Sirisha Senthil, Chandra S. Garudadri.

**Validation:** Aparna Rao, Amiya Pradhan, Sirisha Senthil, Chandra S. Garudadri, P. V. K. S Verma.

**Visualization:** Aparna Rao, Amiya Pradhan.

**Writing – original draft:** Aparna Rao.

**Writing – review & editing:** Aparna Rao, Niranjan Raj, Amiya Pradhan, Sirisha Senthil, Chandra S. Garudadri, P. V. K. S Verma, Prakriti Gupta.

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
