## [Decision Letter · Decision Letter 0]

18 Feb 2020

PONE-D-19-26019

Visual impairment in pseudoexfoliation from four tertiary centres in India

PLOS ONE

Dear Dr. Rao,

Thank you for submitting your manuscript to PLOS ONE. After careful consideration, we feel that it has merit but does not fully meet PLOS ONE’s publication criteria as it currently stands. Therefore, we invite you to submit a revised version of the manuscript that addresses the points raised during the review process.

Two learned reviewers have provided extensive comments and criticisms. The manuscript could be revised incorporating commensurate changes that address all these comments and criticisms. 

We would appreciate receiving your revised manuscript by Apr 03 2020 11:59PM. To enhance the reproducibility of your results, we recommend that if applicable you deposit your laboratory protocols in protocols.io, where a protocol can be assigned its own identifier (DOI) such that it can be cited independently in the future. For instructions see: http://journals.plos.org/plosone/s/submission-guidelines#loc-laboratory-protocols

We look forward to receiving your revised manuscript.

Kind regards,

Sanjoy Bhattacharya

Academic Editor

PLOS ONE

Journal Requirements:

1. Please provide additional details regarding participant consent. In the ethics statement in the Methods and online submission information, please ensure that you have specified (1) whether consent was informed and (2) what type you obtained (for instance, written or verbal, and if verbal, how it was documented and witnessed). If your study included minors, state whether you obtained consent from parents or guardians. If the need for consent was waived by the ethics committee, please include this information.

2) In your Methods section, please provide additional information about the participant recruitment method and the demographic details of your participants. Please ensure you have provided sufficient details to replicate the analyses such as: a) the recruitment date range (month and year), b) a description of any inclusion/exclusion criteria that were applied to participant recruitment, c) a statement as to whether your sample can be considered representative of a larger population and d) a description of how participants were recruited.

3) Please provide a sample size and power calculation in the Methods, or discuss the reasons for not performing one before study initiation. "

4. Thank you for including your ethics statement: Institutional review board of LV Prasad eye institute, MTC campus, Bhubaneswar

IEC number: 2016-12-CB-12

6. Thank you for stating in your Funding Statement:

This work was partly funded by the Wellcome Trust/DBT India  Alliance  grant: ref no IA/CPHI/15/1/502031 awarded to Aparna Rao.

Please include your amended Funding Statement within your cover letter. We will change the online submission form on your behalf

Reviewers' comments:

Reviewer's Responses to Questions

**Comments to the Author**

1. Is the manuscript technically sound, and do the data support the conclusions?

Reviewer #1: Yes

Reviewer #2: Yes

Reviewer #3: Partly

Reviewer #4: Yes

2. Has the statistical analysis been performed appropriately and rigorously? 

Reviewer #1: Yes

Reviewer #2: Yes

Reviewer #3: Yes

Reviewer #4: Yes

3. Have the authors made all data underlying the findings in their manuscript fully available?

Reviewer #1: Yes

Reviewer #2: Yes

Reviewer #3: No

Reviewer #4: Yes

4. Is the manuscript presented in an intelligible fashion and written in standard English?

Reviewer #1: Yes

Reviewer #2: Yes

Reviewer #3: Yes

Reviewer #4: No

5. Review Comments to the Author

Reviewer #1: This Research Article designed Prospectively to analyse the disease burden of pseudoexfoliation stages and the data are supportive, Statistical Analysis performed appropriately, All datas underlying the findings are available, The manuscript is written in standard English.

Reviewer #2: the only thing I recommend is to add a definition of pseudo exfoliation to the article. the article was written in a very good way, understandable language, clear, and no typo in this article. also the tables were well organized and easy to understand by the reader.

Reviewer #3: I read with interest the article entitled .: "Visual impairment in pseudoexfoliation from four tertiary centers in India" by Aparna Rao al., Regarding pseudoexfoliation (PXF) disease stages from East and South India. This research concerns a large number of individuals (over 6.ooo patients). I found it very difficult to read this article because I didn't like it. From an epidemiological point of view this syndrome first discovered in the Scandinavian countries is now considered ubiquitous. The authors state that it is not known why this glaucoma comes. Actually, the furfuraceous material that can be found in any organ indicates a serious alteration of the metabolism of the tissues involved. High pressure glaucoma consists from a molecular point of view in pro-apoptotic signals that reach the head of the optic nerve and induce the death of ganglion cells. Therefore, trabecular meshwork is the first tissue to be involved (see Saccà et al. From DNA damage to functional changes of the trabecular meshwork in aging and glaucoma. Aging Res Rev. 2016; 29: 26-41) and trabecular dandruff there suggests that this structure is seriously damaged and therefore unable to fulfill its barrier functions (Saccà et al. The Outflow Pathway: A Tissue With Morphological and Functional Unity. J Cell Physiol. 2016; 231: 1876-93), hence the increase in intraocular pressure and glaucoma. Even the diagnosis of pseudoexfoliative glaucoma may not be so simple, indeed, according to the only macroscopic presence of the dandruff is uncertain. This syndrome acts like multiple sclerosis. Depending on the affected tissue, the nosological entity of the disease manifests itself: if the iris is affected, iris atrophic changes occur if the lens is affected, the cataract occurs, if the zonula is affected, lens dislocation occurs, if the trabecular meshwork is affected, have glaucoma.

Reviewer #4: Although in general the manuscript is presented in an intelligible fashion, there are some errors in the use and structure of the English language which make it hard to follow at some points. Proofreading and better use of the english language should be highly considered. Some of them are highlighted in the comments.

methods in abstract, Line 37: clinical "and" demographic details instead of clinical, demographic should be used

Line 87: missing the word "primary" before angle closure glaucoma (PACG)

Line 88: focus instead of focuses should be used

Line 111: "electronic media record database": do the authors mean media or medical?

Line 111-116: sentence is hard to follow. medications is written twice.

Line 123: Does "pattern of deposits, diagnosis, ocular associations and number of medications at baseline" refer to data collected from referrals outside? It is not clear to the reader.

Line 137: "open or closed angles on gonioscopy, radial" missing the word "and" before radial

Line 143: Missing the word pseudoexfoliation before glaucoma

Line 161: "anti-glaucoma educations". educations should be replaced by the appropriate word

Line 162: "(for totally bind asymptomatic eyes)"bind should be replaced by the appropriate word

Line 184: "proportions among al patients"al should be replaced by the appropriate word

Line 185: "Since the diagnosis or severity of PXF in one eye has no proven correlation to the risk /condition of the other eye", However the authors should consider that there is a variable but increased probability of developing Pseudoexfoliation in the unaffected eye

Line 191: locations instead of location

Line 198: add "the" before rest

Line 210" authors should include if this finding was statistically significant

Line 212: add "being" before significantly

Line 217: missing word after medical/surgical

Line 220: "were" should be replaced with "was"

Line 226: abbreviation AGM was not explained

Line 256: The authors mention that anterior and posterior segment pathologies were noted and analyzed. However it is not clear If they stratified for these pathologies as they could significantly impact visual acuity apart from the glaucomatous changes.

Line 366: removed one of the two "with"

Line 508:SD was not included in the list with abbreviations

Figure 1 does not show up properly. Please review.

Figure 2 seems blurry.

6. PLOS authors have the option to publish the peer review history of their article (what does this mean?). If published, this will include your full peer review and any attached files.

Reviewer #1: Yes: NAYEF K ALSHAMMARI

Reviewer #2: Yes: Feras Mohder

Reviewer #3: Yes: Sergio C. Saccà

Reviewer #4: No

---

## [Author Response · Author response to Decision Letter 0]

22 Mar 2020

• PONE-D-19-26019

Visual impairment in pseudoexfoliation from four tertiary centres in India

PLOS ONE

Dear Dr. Rao,

Thank you for submitting your manuscript to PLOS ONE. After careful consideration, we feel that it has merit but does not fully meet PLOS ONE’s publication criteria as it currently stands. Therefore, we invite you to submit a revised version of the manuscript that addresses the points raised during the review process.

Two learned reviewers have provided extensive comments and criticisms. The manuscript could be revised incorporating commensurate changes that address all these comments and criticisms. 

We would appreciate receiving your revised manuscript by Apr 03 2020 11:59PM. To enhance the reproducibility of your results, we recommend that if applicable you deposit your laboratory protocols in protocols.io, where a protocol can be assigned its own identifier (DOI) such that it can be cited independently in the future. For instructions see: http://journals.plos.org/plosone/s/submission-guidelines#loc-laboratory-protocols

• A rebuttal letter that responds to each point raised by the academic editor and reviewer(s). This letter should be uploaded as separate file and labeled 'Response to Reviewers'.

• A marked-up copy of your manuscript that highlights changes made to the original version. This file should be uploaded as separate file and labeled 'Revised Manuscript with Track Changes'.

• An unmarked version of your revised paper without tracked changes. This file should be uploaded as separate file and labeled 'Manuscript'.

We look forward to receiving your revised manuscript.

Kind regards,

Sanjoy Bhattacharya

Academic Editor

PLOS ONE

Journal Requirements:

1. Please provide additional details regarding participant consent. In the ethics statement in the Methods and online submission information, please ensure that you have specified (1) whether consent was informed and (2) what type you obtained (for instance, written or verbal, and if verbal, how it was documented and witnessed). If your study included minors, state whether you obtained consent from parents or guardians. If the need for consent was waived by the ethics committee, please include this information.

Answer1: We have made suggested corrections and additions in the text and submission regarding ethics statement, Page 6. 

2) In your Methods section, please provide additional information about the participant recruitment method and the demographic details of your participants. Please ensure you have provided sufficient details to replicate the analyses such as: a) the recruitment date range (month and year), b) a description of any inclusion/exclusion criteria that were applied to participant recruitment, c) a statement as to whether your sample can be considered representative of a larger population and d) a description of how participants were recruited.

Answer: We have now added and expanded the recruitment details of the participants of the study which includes the recruitment date or representative sample details, Page 6. The inclusion/exclusion criteria have already been detailed elaborately in the text in the section clinical definition and stratification, page 7, which has been maintained and elaborated, as suggested by the reviewer. 

3) Please provide a sample size and power calculation in the Methods, or discuss the reasons for not performing one before study initiation. "

Answer3: We have now included a sample size calculation in the methods/statistics section, page 9.

4. Thank you for including your ethics statement: Institutional review board of LV Prasad eye institute, MTC campus, Bhubaneswar

IEC number: 2016-12-CB-12

Answer 4: The ethics statement has been added, as suggested, Page 6.

Answer 5: Additional data has been provided as a separate attachment now

6. Thank you for stating in your Funding Statement:

This work was partly funded by the Wellcome Trust/DBT India Alliance grant: ref no IA/CPHI/15/1/502031 awarded to Aparna Rao.

Please include your amended Funding Statement within your cover letter. We will change the online submission form on your behalf

Answer 6: The funding statement has been amended, as suggested.

Reviewers' comments:

Reviewer's Responses to Questions

Comments to the Author

1. Is the manuscript technically sound, and do the data support the conclusions?

Reviewer #1: Yes

Reviewer #2: Yes

Reviewer #3: Partly

Reviewer #4: Yes

2. Has the statistical analysis been performed appropriately and rigorously?

Reviewer #1: Yes

Reviewer #2: Yes

Reviewer #3: Yes

Reviewer #4: Yes

3. Have the authors made all data underlying the findings in their manuscript fully available?

Reviewer #1: Yes

Reviewer #2: Yes

Reviewer #3: No

Reviewer #4: Yes

4. Is the manuscript presented in an intelligible fashion and written in standard English?

Reviewer #1: Yes

Reviewer #2: Yes

Reviewer #3: Yes

Reviewer #4: No

5. Review Comments to the Author

Reviewer #1: This Research Article designed Prospectively to analyse the disease burden of pseudoexfoliation stages and the data are supportive, Statistical Analysis performed appropriately, All datas underlying the findings are available, The manuscript is written in standard English.

Answer1: We thank the reviewer for the positive comments which encourage us to explore further into pseudoexfoliation which is a baffling disease entity. We are encouraged also to keep up the aulity of our work to meet expectations of the reviewers in the future too.

Reviewer #2: the only thing I recommend is to add a definition of pseudo exfoliation to the article. the article was written in a very good way, understandable language, clear, and no typo in this article. also the tables were well organized and easy to understand by the reader.

Answer 2: We than the reviewer for this suggestion which has been added, as suggested.

Reviewer #3: I read with interest the article entitled .: "Visual impairment in pseudoexfoliation from four tertiary centers in India" by Aparna Rao al., Regarding pseudoexfoliation (PXF) disease stages from East and South India. This research concerns a large number of individuals (over 6.ooo patients). I found it very difficult to read this article because I didn't like it. From an epidemiological point of view this syndrome first discovered in the Scandinavian countries is now considered ubiquitous. The authors state that it is not known why this glaucoma comes. Actually, the furfuraceous material that can be found in any organ indicates a serious alteration of the metabolism of the tissues involved. High pressure glaucoma consists from a molecular point of view in pro-apoptotic signals that reach the head of the optic nerve and induce the 1death of ganglion cells. Therefore, trabecular meshwork is the first tissue to be involved (see Saccà et al. From DNA damage to functional changes of the trabecular meshwork in aging and glaucoma. Aging Res Rev. 2016; 29: 26-41) and trabecular dandruff there suggests that this structure is seriously damaged and therefore unable to fulfill its barrier functions (Saccà et al. The Outflow Pathway: A Tissue With Morphological and Functional Unity. J Cell Physiol. 2016; 231: 1876-93), hence the increase in intraocular pressure and glaucoma. Even the diagnosis of pseudoexfoliative glaucoma may not be so simple, indeed, according to the only macroscopic presence of the dandruff is uncertain. This syndrome acts like multiple sclerosis. Depending on the affected tissue, the nosological entity of the disease manifests itself: if the iris is affected, iris atrophic changes occur if the lens is affected, the cataract occurs, if the zonula is affected, lens dislocation occurs, if the trabecular meshwork is affected, have glaucoma.

Answer 3: We agree with the reviewer’s pertinent views regarding possible mechanisms of tissues getting affected with the exfoliative material. The disease is ubiquitous as pointed out however this occurs with varied prevalance rates across the globe. This topic of glaucoma in PXF is indeed a complex issue which has baffled many scientist and has eluded a clear description of events leading to glaucoma. Yet, it is known that presence of exfoliative material by itself does not mean functional disturbances which is reported in studies showing no correlation of extent of exfoliative deposits in eye with presence of glaucoma (Cobb J et al- Br J Ophthalmol 2004; 88:1002-1003) If material causes mechanical blockage of TM pores with the deposits in all eyes, glaucoma would be seen in all eyes with PXF which is rarely seen in clinical practice. While it is known that these may cause local functional disturbances in the tissues including the TM, the mechanism by which that happens and what triggers the disturbance in some eyes at specific time points only is unknown. This is exactly the reason why all eyes with PXF do not end up with glaucoma even in the same patient with bilateral disease. The molecular events triggering the functional tissue damage is also largely unclear which is discussed in the review article (reference 1). Similar perplexing issues of the disease are discussed in the Journal of glaucoma supplement 1, Volume 27, 2018 discussing several perplexing issues on this disease .Our own lab is working at unraveling the disturbances in eyes with PXF and those without PXG at a molecular level. We also are aware of TM functional damage and cell death in glaucoma in general but would like to clarify that it is not only apoptosis that causes cell death in the human TM in glaucoma. My PhD thesis study evaluating cellular mechanisms of cell death in TM (which largely involved insights from Dr Sacca’s work an articles) found specific molecular events being triggered at specific severity of tissue damage which is not restricted to apoptotic signals only and rather involved a down-regulation of apoptotic molecules contrary to traditional wisdom (paper under review). We are also studying how TM in PXG eyes are affected and are working on an in vitro-model of the same. While several others are also working on the questions the reviewer has suggested and alluded above, we still believe that the glaucoma pathogenesis in PXF remains a mystery. This is the reason why we clinicians cannot predict which eyes are at risk of developing glaucoma and also fall short in predicting when the onset of glaucoma/or tissue damage can occur in any eye. We would be delighted to share our preliminary results with the reviewer when experiments complete regarding the above points of TM damage in PXF and we would be happy to get further insights from the reviewer on this topic. 

Reviewer #4: Although in general the manuscript is presented in an intelligible fashion, there are some errors in the use and structure of the English language which make it hard to follow at some points. Proofreading and better use of the english language should be highly considered. Some of them are highlighted in the comments.

methods in abstract, Line 37: clinical "and" demographic details instead of clinical, demographic should be used

Line 87: missing the word "primary" before angle closure glaucoma (PACG)

Line 88: focus instead of focuses should be used

Line 111: "electronic media record database": do the authors mean media or medical?

Line 111-116: sentence is hard to follow. medications is written twice.

Line 123: Does "pattern of deposits, diagnosis, ocular associations and number of medications at baseline" refer to data collected from referrals outside? It is not clear to the reader.

Line 137: "open or closed angles on gonioscopy, radial" missing the word "and" before radial

Line 143: Missing the word pseudoexfoliation before glaucoma

Line 161: "anti-glaucoma educations". educations should be replaced by the appropriate word

Line 162: "(for totally bind asymptomatic eyes)"bind should be replaced by the appropriate word

Line 184: "proportions among al patients"al should be replaced by the appropriate word

Line 185: "Since the diagnosis or severity of PXF in one eye has no proven correlation to the risk /condition of the other eye", However the authors should consider that there is a variable but increased probability of developing Pseudoexfoliation in the unaffected eye

Line 191: locations instead of location

Line 198: add "the" before rest

Line 210" authors should include if this finding was statistically significant

Line 212: add "being" before significantly

Line 217: missing word after medical/surgical

Line 220: "were" should be replaced with "was"

Line 226: abbreviation AGM was not explained

Line 256: The authors mention that anterior and posterior segment pathologies were noted and analyzed. However it is not clear If they stratified for these pathologies as they could significantly impact visual acuity apart from the glaucomatous changes.

Line 366: removed one of the two "with"

Line 508:SD was not included in the list with abbreviations

Figure 1 does not show up properly. Please review.

Figure 2 seems blurry.

Answer: We than the reviewer for such an exhaustive review and correction at appropriate places. The text has been edited for grammar and typos. All detailed corrections given above have been incorporated, as suggested. The resolution of figures has also been improved, as suggested.

---

## [Decision Letter · Decision Letter 1]

4 May 2020

Visual impairment in pseudoexfoliation from four tertiary centres in India

PONE-D-19-26019R1

Dear Dr. Rao,

We are pleased to inform you that your manuscript has been judged scientifically suitable for publication and will be formally accepted for publication once it complies with all outstanding technical requirements.

With kind regards,

Sanjoy Bhattacharya

Academic Editor

PLOS ONE

Additional Editor Comments (optional):

Reviewers' comments:

Reviewer's Responses to Questions

**Comments to the Author**

1. If the authors have adequately addressed your comments raised in a previous round of review and you feel that this manuscript is now acceptable for publication, you may indicate that here to bypass the “Comments to the Author” section, enter your conflict of interest statement in the “Confidential to Editor” section, and submit your "Accept" recommendation.

Reviewer #1: All comments have been addressed

Reviewer #4: All comments have been addressed

2. Is the manuscript technically sound, and do the data support the conclusions?

Reviewer #1: Yes

Reviewer #4: Yes

3. Has the statistical analysis been performed appropriately and rigorously? 

Reviewer #1: Yes

Reviewer #4: Yes

4. Have the authors made all data underlying the findings in their manuscript fully available?

Reviewer #1: Yes

Reviewer #4: Yes

5. Is the manuscript presented in an intelligible fashion and written in standard English?

Reviewer #1: Yes

Reviewer #4: Yes

6. Review Comments to the Author

Reviewer #1: This Research Article designed Prospectively to analyse the disease burden of pseudoexfoliation stages and the data are supportive, Statistical Analysis performed appropriately, All datas underlying the findings are available, The manuscript is written in standard English.

Reviewer #4: (No Response)

7. PLOS authors have the option to publish the peer review history of their article (what does this mean?). If published, this will include your full peer review and any attached files.

Reviewer #1: Yes: NAYEF K ALSHAMMARI

Reviewer #4: No

---

## [Editor Report · Acceptance letter]

18 May 2020

PONE-D-19-26019R1 

Visual impairment in pseudoexfoliation from four tertiary centres in India 

Dear Dr. Rao:

I am pleased to inform you that your manuscript has been deemed suitable for publication in PLOS ONE. Congratulations! Your manuscript is now with our production department. 

With kind regards,

on behalf of

Dr. Sanjoy Bhattacharya 

Academic Editor

PLOS ONE